# Unified Uncertainty Estimation for Cognitive Diagnosis Models

## ABSTRACT

Cognitive diagnosis models have been widely used in different areas, especially intelligent education, to measure users' proficiency levels on knowledge concepts, based on which users can get personalized instructions. As the measurement is not always reliable due to the weak links of the models and data, the uncertainty of measurement also offers important information for decisions. However, the research on the uncertainty estimation lags behind that on advanced model structures for cognitive diagnosis. Existing approaches have limited inefficiency and leave an academic blank for sophisticated models which have interaction function parameters (e.g., deep learning-based models). To address these problems, we propose a unified uncertainty estimation approach for a wide range of cognitive diagnosis models. Specifically, based on the idea of estimating the posterior distributions of cognitive diagnosis model parameters, we first provide a unified objective function for mini-batch based optimization that can be more efficiently applied to a wide range of models and large datasets. Then, we modify the reparameterization approach in order to adapt to parameters defined on different domains. Furthermore, we decompose the uncertainty of diagnostic parameters into data aspect and model aspect, which better explains the source of uncertainty. Extensive experiments demonstrate that our method is effective and can provide useful insights into the uncertainty of cognitive diagnosis.

## CCS CONCEPTS

• **Applied computing → E-learning**.

## KEYWORDS

Intelligent Education, Cognitive Diagnosis, Uncertainty

**ACM Reference Format:**
Anonymous Author(s). 2018. Unified Uncertainty Estimation for Cognitive Diagnosis Models. In *Proceedings of Make sure to enter the correct conference title from your rights confirmation emai (Conference acronym 'XX)*. ACM, New York, NY, USA, 9 pages. https://doi.org/XXXXXXX.XXXXXXX

## 1 INTRODUCTION

Cognitive diagnosis is a class of methods that have been widely studied in areas such as education [19], psychometric [27], and medical diagnosis [30]. The main purpose of cognitive diagnosis is to obtain examinees' cognitive states from their activities. Particularly, in educational area, such as the online learning platforms, cognitive diagnosis obtains students' knowledge proficiencies from the their

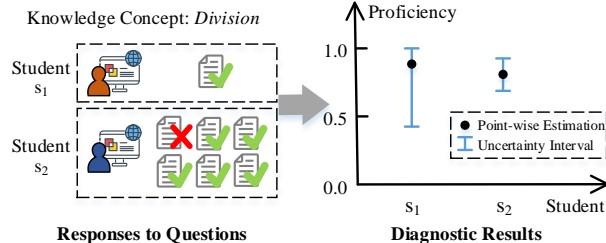

**Figure 1: A toy example.**

learning activities (e.g., question answering), as well as estimates the attributes of questions (e.g., question difficulty). A toy example is illustrated in Figure 1, where two students have answered questions that relate to the knowledge concept "*Division*". After diagnosis, we know that $s_1$ has mastered "*Division*" well while $s_2$ has a lower proficiency (black points). Cognitive diagnosis usually serves as the core of intelligent tutoring systems, which provide personalized support for learners.

In practice, however, the diagnostic results of students are not always highly reliable. In the example of Figure 1, although both students $s_1$ and $s_2$ are diagnosed to have high proficiency of "*Division*", the diagnostic result of $s_1$ is not as reliable as $s_2$. The reason is that $s_1$'s proficiency of "*Division*" is inferred based on a single response related to "*Division*", which may cause severe bias. The uncertainty of diagnosis has important influence on personalized teaching. The system can assign less practice of "*Division*" to $s_2$; while for $s_1$, more questions or better cognitive diagnosis models are needed to obtain an exact proficiency assessment. Furthermore, in a recommender system, more diverse learning resources can be recommended to students with higher uncertainty [13]. In computerized adaptive testing, reducing uncertainty of diagnosis is an important target when selecting the next test question for an examinee [2]. However, most existing diagnosis models cannot tell how confident they are with their point-wise diagnosis.

In recent years, more sophisticated model structures have been proposed for better diagnosis, including deep learning-based models such as NeuralCD [32]. However, the research on the uncertainty estimation of cognitive diagnosis remains on several traditional non-deep learning based models. For example, Bayesian method is the most representative for item response theory (IRT) based models [9]. The application of existing methods is limited due to the following challenges. 1) Limited application range of training algorithms. The widely accepted training algorithms for existing methods, such as Expectation-Maximum (EM) based algorithms and Metropolis-Hasting (MH) sampling based algorithms, are inefficient or even inapplicable to complex diagnosis models (e.g., deep learning-based models) having large scale parameters and on large datasets. 2) Insufficient estimation of parameters. Generally, there are two types of parameters in cognitive diagnosis models, i.e., the diagnostic parameters that represent the features of student and

questions, and function parameters that decide the interaction functions among diagnostic parameters. Existing methods only consider diagnostic parameters, because they are proposed based on traditional cognitive diagnosis models, where the interaction functions are fixed without extra parameters. However, in the state-of-the-art deep learning-based models, the interaction functions are modeled with neural networks, where additional uncertainty from neural network parameters should be considered.

**Our Work.** In this paper, we propose a unified **U**ncertainty estimation approach for **C**ognitive **D**iagnosis models (abbreviated as UCD), which can both be applied to traditional latent trait models and fill the vacancy for deep learning-based models. 1) Based on an idea of learning the posterior distributions of the parameters, we derive a unified objective function for mini-batch based optimization, which can be applied to both deep and non-deep learning models. 2) We propose a derivative reparameterization approach, which not only facilitate the efficient gradient descending-based training, but also adapts to parameters with different domains of definition. 3) By further consideration of the difference between diagnostic parameters and function parameters, we factorize the uncertainty of diagnostic parameters into data uncertainty and model uncertainty. Through extensive experiments on real-world datasets, we validate the effectiveness of UCD and provide some useful insights into the uncertainty of cognitive diagnosis models. The codes and public data are available at: https://anonymous.4open.science/r/UCD-FD2D.

## 2 RELATED WORK

**Cognitive Diagnosis.** Existing cognitive diagnosis methods can be generally classified into non-deep learning models and deep learning-based models. Representative non-deep learning cognitive diagnosis models include continuous latent trait models, such as Item Response Theory (IRT) [9] and Multidimensional Item Response Theory (MIRT) [26]; and discrete classification models, such as Deterministic Input Noisy "And" Gate model [6], and Higher-order DINA [7]. By contrast, deep learning-based approaches achieve state-of-the-art and capture attentions in recent year. Wang et al. [31] proposed a NeuralCD framework that introduces neural networks to learn the interaction between students and questions while keeping interpretability. Several extensions based on NeuralCD have been proposed, such as [20, 23, 32, 34].

**Uncertainty Quantification.** Uncertainty quantification plays a critical role in the process of decision making and optimization in many fields [14, 21]. In cognitive diagnosis, the uncertainty of diagnostic parameters has been studied for traditional models. Fully Bayesian sampling-based methods [25]) and the multiple imputation method [35] characterize the uncertainty of IRT and MIRT by the variations of diagnostic results. Frequentist methods [24, 28] use standard error to reflect the uncertainty. Duck-Mayr et al. [8] proposed a Gaussian process based method for nonparametric IRT models. However, the estimation algorithm could be time consuming, and function parameters are not considered. In deep learning, Bayesian approximation and ensemble learning techniques are two widely-studied types of methods [1] that quantify the uncertainty. Bayesian approximation typically uses a probability distribution to characterize the uncertainty of parameters and model outputs.

Representative methods include the Monte Carlo dropout [33], variational inference [29], and Bayesian neural network based models [3]. Ensemble learning approaches [10, 17] train the deep learning model multiple times and then average the model predictions. Although inspiring, these methods have not been applied to CDMs yet. It should be noted that in CDMs, the focus is the diagnostic results (i.e., the estimated parameters) instead of the model predictions, which is opposite to deep learning models. Moreover, the difference between diagnostic parameters and function parameters are not recognized in existing methods.

## 3 PRELIMINARY
### 3.1 Task Overview

In the educational area, cognitive diagnosis is essentially a measurement of students' knowledge states. Through fitting students' response data by cognitive diagnosis models, the estimated values of student-related parameters are the diagnostic results, which represent the students' levels of knowledge mastery. Suppose there are students $S = \{s_1, s_2, \ldots, s_M\}$, questions $E = \{e_1, e_2, \ldots, e_N\}$, and the Q-matrix $Q \in \{0, 1\}^{N \times K}$ which indicates the related knowledge concepts (KC) of the questions (i.e., $Q_{jk} = 1$ means that question $e_j$ involves knowledge concept $c_k$). Then, the cognitive diagnosis task can be formalized as follows.

*Problem Definition.* The observed data includes students' response logs $R = \{r_{ij}\}$ and the Q-matrix $Q$, where $r_{ij} \in \{0, 1\}$ denotes the student $s_i$'s response to question $e_j$ (i.e., incorrect or correct). Our goal is to estimate the uncertainty of diagnostic results (e.g., students' proficiencies on knowledge concepts) provided by cognitive diagnosis models. Here, the probability distribution is adopted to depict the uncertainty.

### 3.2 Representative Cognitive Diagnosis Models

We briefly introduce the basic structure of cognitive diagnosis models (CDMs) and some representative methods. Generally, a CDM contains two parts: (1) the diagnostic parameters ($\Phi$), indicating the proficiency levels of students ($\alpha_i$) and properties of questions ($\beta_j$); (2) the interaction function about student and question parameters which outputs the probability of correctly answering the question, i.e., $p_{ij} = F(\alpha_i, \beta_j, \Omega)$, where $\Omega$ denotes the parameters of the interaction function. Figure 2 demonstrates the structures of two representative cognitive diagnosis models, i.e., IRT and NeuralCDM. After training the CDM to fit responses, the estimated diagnostic parameters $\alpha_i$ are diagnostic results.

As a representative traditional model, the IRT estimates the interaction function $p_{ij} = 1/\{1 + e^{-1.7 \times \beta_j^{\text{disc}}(\alpha_i - \beta_j^{\text{diff}})}\}$, where $\beta_j^{\text{disc}}$ and $\beta_j^{\text{diff}}$ indicate the discrimination and difficulty of question $e_j$ respectively ($\beta_j = \{\beta_j^{\text{disc}}, \beta_j^{\text{diff}}\}$), and $\alpha_i$ indicates the ability of student $s_i$. IRT has been extended to Multidimensional IRT (MIRT) by using multidimensional vectors of student and question traits [26]. There is no extra functional parameters in these CDMs, i.e., $\Omega = \emptyset$.

As for deep learning-based cognitive diagnosis models, Wang et al. proposed a general framework as well as a model called NeuralCDM, where the interaction function is learned from data by neural networks [31]. The formulation is as follows:

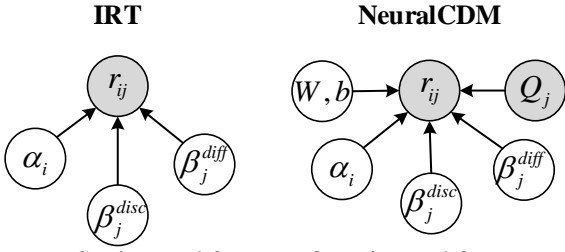

Figure 2: The model structures of IRT and NeuralCDM

$$x_{ij} = Q_j \circ (\alpha_i - \beta_j^{\text{diff}}) \times \beta_j^{\text{disc}}, \quad (1)$$

$$f_1 = \text{Sigmoid}(W_1 \times x_{ij} + b_1), \quad (2)$$

$$f_2 = \text{Sigmoid}(W_2 \times f_1 + b_2), \quad (3)$$

$$p_{ij} = \text{Sigmoid}(W_3 \times f_2 + b_3), \quad (4)$$

where $\alpha_i$ indicates student $s_i$'s proficiency on each knowledge concept; $\beta_j^{\text{diff}}$ indicates the difficulty of each knowledge concept tested by question $e_j$; $\beta_j^{\text{disc}}$ indicates the discrimination of question $e_j$; $Q_j$ is the j-th row of Q-matrix. $\Omega = \{W_1, W_2, W_3, b_1, b_2, b_3\}$ are network parameters, where each element in $W_*(* = 1, 2, 3)$ is nonnegative.

## 4 UNCERTAINTY ESTIMATION FOR COGNITIVE DIAGNOSIS MODELS

We first introduce an overview of our approach. Then, we provide a unified objective function for mini-batch based training which can be applied to different CDMs on large datasets, and the reparameterization trick that facilitates the gradient computation of different parameter distributions. Finally, we introduce decomposition of the uncertainty to better estimate the parameters.

### 4.1 Overview

As most continuous latent trait CDMs and existing deep learning-based CDMs fall under the umbrella of the framework described in 3.2, we choose to make minor modification of the framework so that our approach can be applied to a wider range of CDMs and avoid impairing the diagnosing ability of the original model structures. Furthermore, in order to obtain the uncertainty of parameters during model training, we change the point-wise estimations of parameters into estimating the posterior distributions. The variance of a posterior distribution directly depicts the uncertainty of the parameter. Uncertainty intervals can also be obtained as an indicator of the uncertainty, which is adopted by some studies [5, 11]. Consequently, we propose a unified Bayesian approach called UCD.

For convenience, we treat the parameters as random variables and represent all the variables with $\Psi = \Phi \cup \Omega$, where $\Phi$ denotes the diagnostic variables, including student variables $\alpha = \{\alpha_i, i = 1, 2, \ldots, M\}$ and question variables $\beta = \{\beta_j, j = 1, 2, \ldots, N\}$. The overall generative process of the responses $R = \{r_{ij}\}$ modeled by UCD is depicted in Figure 3. To directly estimate the posterior distribution $p(\Psi|R)$ is intractable. Instead, we adopt a practical solution that approximates $p(\Psi|R)$ with a parametric distribution $q(\Psi|\theta)$ which has good statistical properties [3]. Furthermore, by assuming the independence among the variables, the distribution can be factorized to:

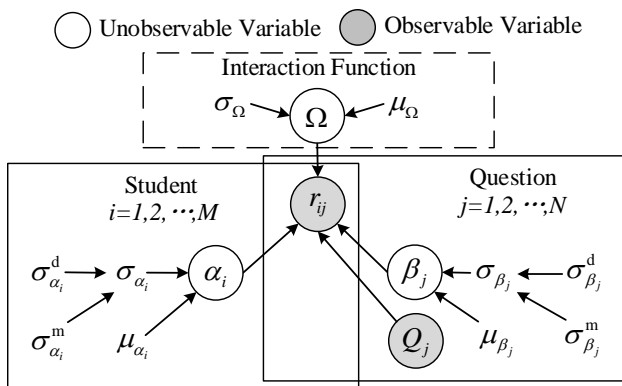

Figure 3: The graphic model of UCD

$$p(\Psi|R) \simeq q(\Psi|\theta) = q(\Phi|\theta_\Phi)q(\Omega|\theta_\Omega), \quad (5)$$

where $\theta_\Phi$ and $\theta_\Omega$ are learnable parameters (notations without circles in Figure 3) that define the distributions of $\Phi$ and $\Omega$ respectively. Therefore, the goal of model training changes to finding the optimal parameters $\theta = \theta_\Phi \cup \theta_\Omega$ that make $q(\Psi|\theta)$ closest to $p(\Psi|R)$. Along this way, we introduce the derivation of the objection function in the following subsection.

### 4.2 Objective Function

In this subsection, we derive the objective function for mini-batch based optimization, which can be used for different CDMs. Primarily, we choose to minimize the Kullback-Leibler divergence ($D_{KL}$) [16], which is a widely accepted measurement of the distance between probability distributions. Therefore, the optimal $\theta^*$ can be calculated as:

$$\theta^* = \arg\min_\theta D_{KL}[q(\Psi|\theta)||p(\Psi|R)]$$
$$= \arg\min_\theta D_{KL}[q(\Psi|\theta)||p(\Psi)] - \mathbb{E}_{q(\Psi|\theta)} \log p(R|\Psi), \quad (6)$$

where $p(\Psi)$ is the prior distribution of the variables. $D_{KL}[q(\Psi|\theta)||p(\Psi)] - \mathbb{E}_{q(\Psi|\theta)} \log p(R|\Psi)$ is not an ideal objective function yet, as there is the calculation of expectation. Based on the Monte Carlo approach [3], the expectation can be approximated with the average of samplings. In addition, we incorporate the mini-batch based training strategy in order to facilitate complicated CDMs and large datasets. Specifically, assuming that there are $M_b$ mini-batches, and for each data sample, we draw $M_c$ variable samples from the distribution $q(\Psi|\theta)$. Then for i-th batch, let $F'_i(\theta) = \pi_i L_A - L_B$, where:

$$L_A = D_{KL}[q(\Psi|\theta)||p(\Psi)], \quad L_B = \sum_j \frac{1}{M_c} \sum_{m=1}^{M_C} \log p(R_j|\Psi_{jm}). \quad (7)$$

Here, $R_j$ is the j-th response in the batch, $\Psi_{jm}$ is the m-th sample from $P(\Psi|\theta)$ for $R_j$, and $\sum_{i=1}^{M_b} \pi_i = 1$. We can adopt $\pi_i = \frac{2^{M_b - i}}{2^{M_b} - 1}$ [3]. Furthermore, we place weights on the KL divergence of diagnostic variables and function variables to adjust their learning rates:

$$L'_A = \zeta_0 D_{KL}[q(\Phi|\theta_\Phi)||p(\Phi)] + \zeta_1 D_{KL}[q(\Omega|\theta_\Omega)||p(\Omega)], \quad (8)$$

where $\zeta_0$ and $\zeta_1$ are hyper-parameters. Finally, the objective function for the i-th mini-batch is:

$$F_i(\theta) = \pi_i L'_A - L_B. \quad (9)$$

Minimizing $F_i(\theta)$ means better approximating the prior distribution (lower $L'_A$) and higher probability of reconstructing the responses (higher $L_B$).

## 4.3 Reparameterization

We adopt gradient descent algorithm to optimize the parameters, as gradient descent can be applied to both deep learning and non-deep learning models, and is more efficient than EM-based or MH sampling-based algorithms in traditional approaches. However, there still exists a problem that, if we directly sample $\Psi$ from the distribution $q(\Psi|\theta)$, the gradient of $\theta$ in $L_B$ will not be able to calculated. Therefore, reparameterization trick is adopted. To facilitate variables defined on different domains and simplify the sampling process, we propose a theorem derived from the proposition in [3] as follows:

THEOREM 4.1. *Suppose there is a function $h(x)$ and its inverse function $g(x)$. Let $\epsilon$ be a random variable having a probability density $\epsilon \sim N(0, 1)$, and let $\Psi = g(\mu + \sigma\epsilon)$. Then we have $h(\Psi) \sim N(\mu, \sigma^2)$, and for a function $f(\Psi, \theta)$, we have:*

$$\frac{\partial}{\partial\theta}\mathbb{E}_{q(\Psi|\theta)}[f(\Psi, \theta)] = \mathbb{E}_{q(\epsilon)}\Big[\frac{\partial f(\Psi, \theta)}{\partial\Psi}\frac{\partial\Psi}{\theta} + \frac{f(\Psi, \theta)}{\partial\theta}\Big]. \quad (10)$$

PROOF. As $g(x)$ is the inverse function of $h(x)$, it is easy to get $h(\Psi) = (\mu + \sigma\epsilon) \sim N(\mu, \sigma^2)$.

Then, we prove that $q(\Psi|\theta)\mathrm{d}\Psi = q(\epsilon)\mathrm{d}\epsilon$.

$$\begin{aligned}
q(\Psi|\theta)\mathrm{d}\Psi &= h'(\Psi)f(h(\Psi))\mathrm{d}\Psi \\
&= h'(\Psi)f(h(\Psi))\mathrm{d}g(h(\Psi)) \\
&= h'(\Psi)f(h(\Psi))g'(h(\Psi))\mathrm{d}h(\Psi) \\
&= f(\mu + \sigma\epsilon)\mathrm{d}(\mu + \sigma\epsilon) \\
&= \frac{1}{\sqrt{2\pi}\sigma}e^{-\frac{(\mu+\sigma\epsilon-\mu)^2}{2\sigma^2}} \cdot \sigma\mathrm{d}\epsilon \\
&= \frac{1}{\sqrt{2\pi}}e^{-\frac{\epsilon^2}{2}}\mathrm{d}\epsilon \\
&= q(\epsilon)\mathrm{d}\epsilon.
\end{aligned}$$

Therefore, we have:

$$\begin{aligned}
\frac{\partial}{\partial\theta}\mathbb{E}_{q(\Psi|\theta)}[f(\Psi, \theta)] &= \frac{\partial}{\partial\theta}\int f(\Psi, \theta)q(\Psi|\theta)\mathrm{d}\Psi \\
&= \frac{\partial}{\partial\theta}\int f(\Psi, \theta)q(\epsilon)\mathrm{d}\epsilon \\
&= \mathbb{E}_{q(\epsilon)}\Big[\frac{\partial f(\Psi, \theta)}{\partial\Psi}\frac{\partial\Psi}{\theta} + \frac{f(\Psi, \theta)}{\partial\theta}\Big].
\end{aligned}$$

$\square$

Based on Theorem 4.1, the partial derivative with respect to $\theta$ of an expectation can be calculated as the expectation of a partial derivative, and the expectation can be further approximated with MC sampling. If we select a distribution for $\Psi$ that $h(\Psi) \sim N(\mu, \sigma^2)$, here $\theta = \{\mu, \sigma\}$, then an unbiased partial derivative with respect to $\theta$ of $\mathbb{E}_{q(\Psi|\theta)}\log p(R|\Psi)$ (in Eq. (6)) can be calculated with the following steps: (1) draw samples of $\epsilon$ from $N(0, 1)$; (2) let $\Psi = \Psi(\theta, \epsilon) = g(\mu + \sigma\epsilon)$; (3) calculate $\partial\mathbb{E}_{q(\Psi|\theta)}\log p(R|\Psi)/\partial\theta = \partial L_B/\partial\theta$.

According to the domain of definition of the $\Psi$, different distributions $q(\Psi|\theta)$ can be selected. Using $\psi$ to denote any variable in $\Psi$, the corresponding distribution can be selected as shown in Table 1.

With the usage of the above probability distributions for each variable $\psi$, the corresponding parameters that need to be estimated during training are $\theta_\psi = \{\mu_\psi, \sigma_\psi\}$, where $\psi \in \Psi = \alpha \cup \beta \cup \Omega$. It should be noted that, with the assumption of variable independence, all variables are fully factorized, i.e., the covariance of a multidimensional variable is 0.

## 4.4 Decomposition of the Uncertainty

A significant difference between diagnostic variables and function variables is that: function variables are affected by all the responses in data, while the diagnostic variables are mainly affected by related responses. For example, in IRT, the distribution of $\alpha_i$ is estimated according to student $s_i$'s responses; in NeuralCDM, the distribution of student $s_i$'s proficiency on knowledge concept $c_k$ ($\alpha_{ik}$) is estimated according to $s_i$'s responses to questions that involve $c_k$. Therefore, even if the responses to a student/question are highly consistent (illustrated as $s_1$ in Figure 1), there still exists relatively high uncertainty if related responses are too few. Accordingly, we decompose the uncertainty of diagnostic variables into model uncertainty and data uncertainty.

To be specific, the distribution parameter $\sigma_\phi$ are decomposed into $\sigma_\phi^m$ and $\sigma_\phi^d$ ($\phi \in \alpha \cup \beta$), where $\sigma_\phi^m$ indicates the uncertainty learned from the CDM, and $\sigma_\phi^d$ is monotonically decreasing with the amount of related responses. In addition, considering that $\sigma_\phi^d$ should be positive and has a diminishing marginal utility when there are sufficiently large number of relevant responses, we formulate it as $\sigma_\phi^d = \lambda_0 e^{-\lambda_1\tau}$, where $\tau$ is the number of responses related to $\phi$; $\lambda_0$ and $\lambda_1$ are learnable weights that adjust the rate of decreasing (two sets of $\lambda_0$ and $\lambda_1$ can be used for questions and students respectively when the amount of responses related to a question differs too much from that to a student). Then, we use $\sigma_\phi = \sigma_\phi^m \times \sigma_\phi^d$.

The whole graphical model of UCD is illustrated in Figure 3, and the training algorithm is summarized in Algorithm 1, where $l$ is the learning rate. Existing gradient descent algorithms, such as SGD [4] and Adam [15], can be adopted to update the parameters (line 11-12). (We will release our code after acceptance.)

## 4.5 Model Complexity

The space complexity of UCD integrated CDMs depends on the $q(\Psi|\theta)$ we choose. In our case, although UCD doubles the number of parameters, the space complexity is still O(M + N + U), where M, N and U are the numbers of students, questions and function parameters.

The increase of the amount of parameters does not affect the time cost much, as it does not change the gradient descent algorithm (we did not observe appreciably more epochs before convergence in our experiments). The extra time cost mainly comes from the sampling process, especially the sampling of neural network parameters. For example, in Eq. (2), $W_1$ is sampled for each data sample in $x_{ij}$, changing the matrix-matrix multiplication ($W_1 \times x_{ij}$) to multiple matrix-vector multiplications, which is difficult for parallel GPU computing. Nevertheless, this is an acceptable trade-off in order to

**Table 1: Distributions selected for variables defined on different domains.**

| Domain of $\psi$ | $h_\psi(x)$ | $g_\psi(x)$ | Examples |
|---|---|---|---|
| $(-\infty, +\infty)$ | $x$ | $x$ | The student ability and question difficulty in IRT and MIRT; the network bias in NeuralCDM. We get $\psi \sim N(\mu, \sigma^2)$. |
| $(a, +\infty)$ | $\ln(x-a)$ | $e^x + a$ | The discrimination in IRT and MIRT; the weights of neural networks in NeuralCDM. Here, $a = 0$, which means $\psi \sim log-norm(\mu, \sigma^2)$. |
| $(a, b)$ | $\text{Logit}(\frac{x-a}{b-a})$ | $\text{Sigmoid}(x)(b-a)+a$ | the student ability, question difficulty and discrimination in Neural-CDM. Here, $a = 0, b = 1$, which means $\psi \sim logit-norm(\mu, \sigma^2)$. |

---

**Algorithm 1** UCD training algorithm

**Input**: Responses R; Q-matrix Q

**Parameter**: Parameters of the approximated posterior distributions, i.e., $\theta_\Phi = \{\mu_\Phi, \sigma_\Phi^m, \lambda_0, \lambda_1\}, \theta_\Omega = \{\mu_\Omega, \sigma_\Omega\}$

**Output**: Approximated posterior distributions of diagnostic variables $q(\Phi|\theta_\Phi)$

1: **while** not converged **do**
2:    **for** batch i in R **do**
3:      **for** variable $\psi$ in $\Phi \cup \Omega$ **do**
4:        Draw $M_c$ samples of $\epsilon$ from $N(0,1)$
5:        **if** $\psi$ is a diagnostic variable in $\Phi$ **then**
6:          $\sigma_\psi^d = \lambda_0 e^{-\lambda_1 \tau}, \sigma_\psi = \sigma_\psi^d \times \sigma_\psi^m$
7:        **end if**
8:        Let $\psi = g_\psi(\mu_\psi + \sigma_\psi \epsilon)$ (Table 1)
9:      **end for**
10:      Calculate the loss $F_i(\theta) = \pi_i L'_A - L_B$, where $L'_A = \zeta_0 D_{KL}[q(\Phi|\theta_\Phi)] + \zeta_1 D_{KL}[q(\Omega|\theta_\Omega)]$, $L_B = \sum_j \frac{1}{M_c} \sum_{m=1}^{M_C} \log p(R_j|\Psi_{jm})$. Eq. ((7)-(9))
11:      **for** $\theta \in \theta_\Phi \cup \theta_\Omega$ **do**
12:        $\theta \leftarrow \theta - l \, \nabla_\theta F_i(\theta)$
13:      **end for**
14:    **end for**
15: **end while**
16: **return** $q(\Phi|\theta_\Phi)$

---

obtain the uncertainty of CDMs, especially in deep learning-based CDMs where traditional uncertainty estimation methods can not be applied.

## 5 EXPERIMENTS

We conduct comprehensive experiments to answer the following research questions:

**RQ1** Can UCD provide reasonable uncertainty for different CDMs?
**RQ2** Whether the captured uncertainty relevant to the decomposed sources?
**RQ3** Can UCD more efficiently deal with sophisticated CDMs and large datasets?
**RQ4** What personalized diagnostic information can UCD provide?
**RQ5** Does UCD avoid impairing the diagnostic ability of the CDMs?

**Table 2: The statistics of the datasets.**

| | FrcSub | Math | Eedi |
|---|---|---|---|
| number of students | 536 | 7,756 | 17,740 |
| number of questions | 20 | 1,993 | 8,987 |
| number of KCs | 8 | 305 | 286 |
| number of responses | 10,720 | 637,798 | 610,032 |

### 5.1 Dataset Description

We use three real-world datasets, i.e., FrcSub, Math and Eedi, in the experiments. FrcSub is a widely used dataset in cognitive diagnosis modeling, which consists of students' responses to fraction-subtraction questions [22]. Math is a dataset collecting the test performances of senior-high school students. Eedi is the dataset released by the NeurIPS 2020 education challenge (track 1), containing students' answers to mathematics questions from Eedi[1]. We use a subset of the original data starting from 04/01/2020 to 05/01/2020. Table 2 shows some basic statistics.

### 5.2 Experimental Setup

To evaluate the effectiveness of our method, we applied UCD to two representative non-deep learning CDMs, i.e., IRT [9] and MIRT [26], and two representative deep learning based CDMs, i,e, NeuralCDM [31] and KaNCD [32]. In addition, we also compare our UCD with the fully Bayesian sampling based method [25] (FB) on IRT and MIRT, multiple imputation [35] (MI) on IRT [2], and the nonparametric method GPIRT [8]. As ensemble based method is the only available baseline that can be directly applied on NeuralCDM and KaNCD, we compare UCD with deep ensemble [17] (DE).

The fully Bayesian sampling-based approach was implemented using PyStan[3] of which the underlying implementation is in C language, and the number of warm up samples is set to 500; GPIRT is implemented based on the R package provided by the authors[4]; the other approaches were implemented with Pytorch[5] in Python. All experiments were run on a Linux server with Intel Xeon Gold 5218 CPU and Tesla V100 GPU.

The responses of each student in the datasets are divided into train:validate:test = 0.7:0.1:0.2. $M_c$ is set to be 5. $\zeta_0$ and $\zeta_1$ are both

---

[1]https://competitions.codalab.org/competitions/25449
[2]Frequentist methods are not compared with because they use standard error instead of probability distribution (or uncertainty interval) to represent the uncertainty of student proficiency.
[3]https://pystan.readthedocs.io/
[4]https://github.com/duckmayr/gpirt/blob/main/
[5]https://pytorch.org/

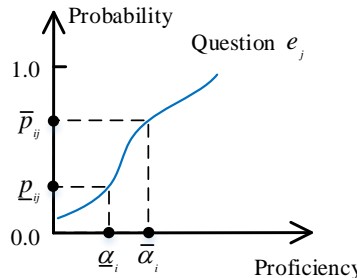

Figure 4: A unidimensional illustration of interval transformation.

selected from [0.01, 0.1, 1, 1.5]. For all the standard deviation parameters ($\sigma_*$), to ensure that they are positive, we instead make $\sigma_* = \text{Softplus}(\eta_*)$, and learn $\eta_*$ through training. We select $N(0, 1)$, log-norm(0,1) and logit-norm(0,1) as the prior distributions for variables defined on $(-\infty, +\infty)$, $(0, +\infty)$ and $(0, 1)$ respectively. To initialize the network variables, we first initialize a matrix $W$ with Kaiming initialization [12], and then let $\mu_W = \ln(|W|)$. The Adam algorithm [15] is used for optimization, and the learning rate is 0.002.

### 5.3 Evaluation of Uncertainty Intervals (RQ1)

The uncertainty of the diagnostic results (i.e., student variable $\alpha$) is characterized by their estimated posterior distributions, and can be further concretized with the confidence intervals (uncertainty intervals) of the distributions. To facilitate the evaluation with observable responses, we project the intervals of students' knowledge proficiencies $[\underline{\alpha}_i, \overline{\alpha}_i]$ to the intervals of model predictions $[\underline{p}_{ij}, \overline{p}_{ij}]$. This is achieved by taking advantage of the monotonicity of CDMs. As the monotonicity assumption in CDMs indicates, the model prediction monotonically increases with any dimension of knowledge proficiency $\alpha_i$ [26]. Figure 4 illustrates a unidimensional example, where the curve depicts the predicted probability (that a student can correctly answer the question $e_j$) with respect to the student's knowledge proficiency. Specifically, we first obtain the 95% confidence interval of the estimated knowledge proficiency $[\underline{\alpha}_i, \overline{\alpha}_i]$, where $\underline{\alpha}_i = g(\mu_{\alpha_i} - 1.96\sigma_{\alpha_i})$ and $\overline{\alpha}_i = g(\mu_{\alpha_i} + 1.96\sigma_{\alpha_i})$. Here, $g(\cdot)$ is the function discussed in Table 1. Next, we sample question variables ($\beta_j$) and network variables ($\Omega$) 50 times and calculate their corresponding predictions with $\underline{\alpha}_i$ and the corresponding interaction of the CDM. $\underline{p}_{ij} = \mathbb{E}_{q(\beta_j, \Omega | \theta_{\beta_j}, \theta_\Omega)} p(r_{ij} = 1 | \underline{\alpha}_i)$ is finally approximated with the average of these predictions. Similarly, $\overline{p}_{ij}$ can be obtained. DE is exceptional, for which $[\underline{\alpha}_i, \overline{\alpha}_i]$ is directly obtained from the predictions of multiple trained CDM instances.

In order to evaluate whether reasonable uncertainty intervals are obtained, Prediction Interval Coverage Probability (PICP) and Prediction Interval Average Width (PIAW) are widely accepted metrics [1]. PICP calculates the proportion of true values lying in the interval, while PIAW calculates the average widths of the intervals. To adapt to binary response labels (0 or 1) in our experiments, we adjust the formulation as follows:

$$PICP = \frac{1}{n} \sum_{i,j} c_{ij}, \quad PIAW = \frac{1}{n} \sum_{i,j} (\overline{p}_{ij} - \underline{p}_{ij}), \quad (11)$$

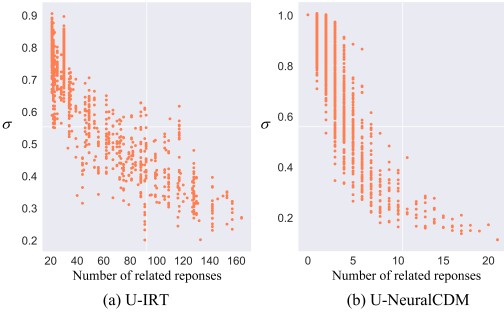

(a) U-IRT  (b) U-NeuralCDM

Figure 5: The $\sigma_\alpha$ of students estimated by U-IRT and U-NCDM.

where $n$ is the number of responses in the test set, and

$$c_{ij} = \begin{cases} 1, & [0.5r_{ij}, 0.5(1 + r_{ij})] \cap [\underline{p}_{ij}, \overline{p}_{ij}] \neq \varnothing, \\ 0, & \text{otherwise.} \end{cases} \quad (12)$$

Well estimated intervals should have a PICP close to the confidence level, and the same PICP with a smaller PIAW indicates a tighter interval. Furthermore, with a certain confidence level, a CDM having a smaller PIAW usually indicates more confident diagnostic results.

The results of the models are presented in Table 3. [6] We have the following observations. First, UCD achieves PICPs closer to 0.95, which indicates accurate uncertainty estimation. Second, on IRT, MI tends to underestimate the uncertainty. The uncertainty estimated by UCD is consistent with the traditional FB method, and UCD performs better than FB on FrcSub and Math. On MIRT, FB overestimates the uncertainty (the abnormally high PIAW), while UCD provides reasonable results. These validate the effectiveness of UCD on traditional CDMs. On NeuralCDM and KaNCD, UCD gets better results most time. Moreover, the comparability issue among the CDMs trained multiple times (i.e., scale linking [18]) is dismissed in DE.

### 5.4 Analysis of the Uncertainty Source (RQ2)

As stated in subsection 4.4, the uncertainty diagnostic variables comes from both data aspect and model aspect. Better insights into the uncertainty source can be useful in applications, such as deciding the number of questions or repeats of knowledge concepts in an examination, and selecting suitable CDMs that have better balance between diagnosis accuracy and model uncertainty on the data. For better understanding, we visualize the uncertainty parameters $\sigma_\alpha$ estimated on Math in Figure 5. For brevity, we use a prefix "U-" to identify the CDMs integrated with UCD.

**For data aspect**, as can be observed in Figure 5, there is a tendency that diagnostic variables with more related responses should have lower uncertainty. To fully validate whether this tendency is captured by UCD, we calculate the Spearman rank correlation coefficient [36] between the $\sigma_\alpha$ of students and the number of responses related to $\alpha$ in training set. The results are presented in Table 4. As expected, we can observe strong negative correlations, which

---

[6]The results of GPIRT on Math and Eedi were not obtained because the iteration stop condition is too hard to meet for these large datasets, causing unacceptable running time.

**Table 3: Experimental results of student performance prediction (uncertainty interval).**

| Dataset | Metric | IRT | | | | MIRT | | NeuralCDM | | KaCND | |
|---|---|---|---|---|---|---|---|---|---|---|---|
| | | FB | MI | GPIRT | UCD | FB | UCD | DE | UCD | DE | UCD |
| FrcSub | PICP | 0.935 | 0.885 | 0.933 | 0.957 | 1.000 | 0.922 | 0.868 | 0.956 | 0.899 | 0.918 |
| | PIAW | 0.340 | 0.277 | 0.335 | 0.342 | 0.999 | 0.264 | 0.085 | 0.472 | 0.191 | 0.247 |
| Math | PICP | 0.867 | 0.802 | - | 0.883 | 1.000 | 0.940 | 0.816 | 0.927 | 0.875 | 0.837 |
| | PIAW | 0.159 | 0.269 | - | 0.194 | 0.990 | 0.393 | 0.084 | 0.468 | 0.176 | 0.143 |
| Eedi | PICP | 0.898 | 0.830 | - | 0.892 | 1.000 | 0.941 | 0.831 | 0.946 | 0.864 | 0.827 |
| | PIAW | 0.266 | 0.226 | - | 0.247 | 0.999 | 0.493 | 0.124 | 0.471 | 0.195 | 0.107 |

**Table 4: The Spearman rank correlations between $\sigma_\alpha$ and the number of related questions. The results of U-IRT and U-MIRT cannot be calculated on FrcSub because all students answer the same number of questions.**

| Dataset | U-IRT | U-MIRT | U-NeuralCDM | U-KaNCD |
|---|---|---|---|---|
| FrcSub | - | - | -0.96 | -0.92 |
| Math | -0.91 | -0.89 | -0.94 | -0.60 |
| Eedi | -0.91 | -0.69 | -0.85 | -0.42 |

**Table 5: The Spearman rank correlations between $\sigma_\alpha^m$ and the fitting ability. The results of U-IRT and U-MIRT cannot be calculated on FrcSub because all students answer the same number of questions.**

| Dataset | U-IRT | U-MIRT | U-NeuralCDM | U-KaNCD |
|---|---|---|---|---|
| FrcSub | - | - | 0.82 | 0.23 |
| Math | 0.73 | 0.53 | 0.90 | 0.05 |
| Eedi | 0.60 | -0.63 | 0.99 | -0.16 |

validate the tendency. Here we focus on the diagnosed proficiencies of students ($\alpha$), which is the goal of CDMs, and same results can be observed for question parameters ($\beta$).

**For model aspect**, as we can observe from Figure 5, although there is a decreasing tendency with the number of responses, there are variances on a certain number of responses, which are cause by $\sigma_\phi^m$. The estimated $\sigma_\phi^m$ has a more complicated relation with the properties of CDMs, which can be difficult to fully analyze. We here provide a viewpoint that we observed in experiments. In general, the distance between model predictions and the true response labels indicates the ability of the model to reconstruct the responses. Therefore, this distance can be an indicator of the model characteristic, which may be relevant to the model uncertainty. Along this way, we calculate the Spearman rank correlation between this distance and the estimated $\sigma_\phi^m$ of student variables.

For CDMs diagnosing latent abilities (no corresponding relationship with Q-matrix, e.g., U-IRT, U-MIRT), the overall distance of student $s_i$ is:

$$\text{dist}(s_i) = \sum_{r_{ij} \in R_i} |\hat{p}_{ij} - r_{ij}|, \tag{13}$$

where $R_i$ is the set of responses of $s_i$ in data; $\hat{p}_{ij}$ is expected prediction of input $s_i$ and $e_j$; $r_{ij}$ is the true response. Then, the Spearman rank correlation between $\{\text{dist}(s_i), i = 1, 2, \ldots, M\}$ and $\{\sigma_{\alpha_i}^m, i = 1, 2, \ldots, M\}$ is calculated.

For CDMs diagnosing explicit knowledge proficiencies (having corresponding relationship with Q-matrix, e.g., U-NeuralCDM, U-KaNCD), the overall distance of student $s_i$'s proficiency on knowledge concept $c_k$ is:

$$\text{dist}(s_i^k) = \sum_{r_{ij} \in R_i^k} |\hat{p}_{ij} - r_{ij}|, \tag{14}$$

where $R_i^k$ is the set of responses of $s_i$ to the questions requiring $c_k$. Then, the Spearman rank correlation between $\{\text{dist}(s_i^k), i = 1, 2, \ldots, M, k = 1, 2, \ldots, K\}$ and $\{\sigma_{\alpha_{ik}}^m, i = 1, 2, \ldots, M, k = 1, 2, \ldots, K\}$

is calculated. The results are presented in Table 5, where we can observe obvious correlations on most models, which partially explains the differences of model uncertainty ($\sigma_\alpha^m$). The relatively weak correlation presented by U-KaNCD should be caused by that KaNCD actually models the associations among knowledge concepts, which is not measured by Eq. (14). It should be noticed that the evaluation here provide a viewpoint to understand $\sigma_\alpha^m$. The whole relation between $\sigma_\alpha^m$ and CDMs can be more complicated.

## 5.5 Comparison of the Efficiency (RQ3)

As stated in the Introduction, one of the limitations of traditional uncertainty estimation approaches is the limited application range of training methods, which are inefficient and even inapplicable to complex cognitive diagnosis models (CDMs) and large datasets. Here, we provide the training time costs (until convergence) of fully Bayesian sampling-based approach, multiple imputation approach, and our UCD in Table 6. We can observe from the table that the model complexity and data size have significant impact on the time cost of traditional approaches. Specifically, FB-MIRT requires much more time cost than FB-IRT, and their time cost increases dramatically on larger dataset, i.e, Math and Eedi. Similarly, GPIRT requires unacceptable time cost when applied on Math and Eedi. In contrast, the time cost increment of UCD is more moderate. Moreover, UCD can be applied to deep learning-based CDMs (e.g., NeuralCDM and KaNCD) where traditional approaches are not applicable. The ensemble based uncertainty estimation approaches from deep learning academia are essentially not for CDMs, and the time cost is N times the original CDM, where N is the number of trials for CDM training. larger N can provide more accurate estimation, but also leads to higher time cost.

## 5.6 Illustration of Diagnostic Information (R4)

Through integrating UCD, a CDM can provide more information about the diagnostic results. Here we present an example of diagnostic results provided by IRT, U-IRT, NeuralCDM and U-NeuralCDM,

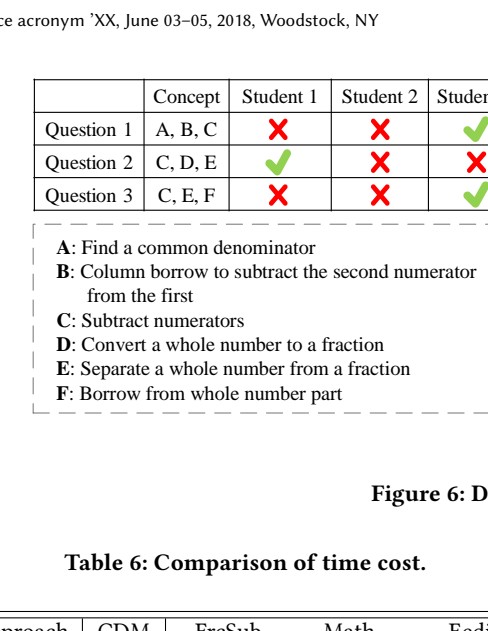

|  | Concept | Student 1 | Student 2 | Student 3 |
|---|---|---|---|---|
| Question 1 | A, B, C | ✗ | ✗ | ✓ |
| Question 2 | C, D, E | ✓ | ✗ | ✗ |
| Question 3 | C, E, F | ✗ | ✗ | ✓ |

**A**: Find a common denominator
**B**: Column borrow to subtract the second numerator from the first
**C**: Subtract numerators
**D**: Convert a whole number to a fraction
**E**: Separate a whole number from a fraction
**F**: Borrow from whole number part

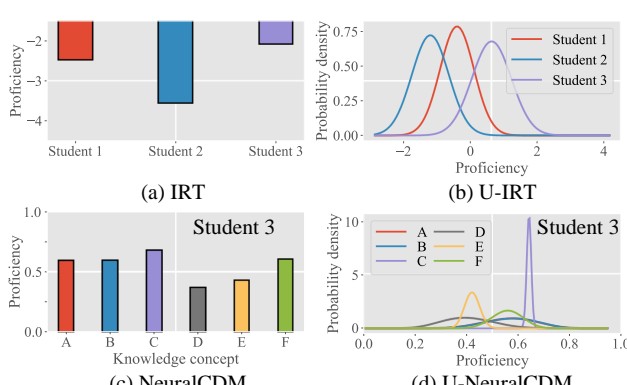

(a) IRT

(b) U-IRT

(c) NeuralCDM

(d) U-NeuralCDM

Figure 6: Differences of diagnostic results.

Table 6: Comparison of time cost.

| Approach | CDM | FrcSub | Math | Eedi |
|---|---|---|---|---|
| FB | IRT | 15s | 1h 21min | 2h 10min |
|  | MIRT | 3min 15s | >12h | >12h |
| MI | IRT | 16min 30s | >12h | >12h |
| GPIRT | IRT | 20s | - | - |
| UCD | IRT | 90s | 7min | 9min 20s |
|  | MIRT | 1min 58s | 7min 45s | 19min 31s |

Table 7: Experimental results of student performance prediction (point-wise/expectation).

| Dataset | FrcSub | | Math | | Eedi | |
|---|---|---|---|---|---|---|
| Metric | AUC | Acc | AUC | Acc | AUC | Acc |
| IRT | 0.829 | 0.778 | 0.809 | 0.779 | 0.796 | 0.758 |
| U-IRT | 0.881 | 0.805 | 0.815 | 0.781 | 0.808 | 0.766 |
| MIRT | 0.877 | 0.807 | 0.810 | 0.774 | 0.781 | 0.744 |
| U-MIRT | 0.894 | 0.822 | 0.822 | 0.782 | 0.806 | 0.764 |
| NeuralCDM | 0.894 | 0.824 | 0.808 | 0.772 | 0.811 | 0.768 |
| U-NeuralCDM | 0.899 | 0.826 | 0.806 | 0.775 | 0.810 | 0.765 |
| KaNCD | 0.900 | 0.835 | 0.824 | 0.783 | 0.809 | 0.765 |
| U-KaNCD | 0.903 | 0.838 | 0.822 | 0.783 | 0.811 | 0.764 |

in Figure 6. We randomly select three students from FrcSub, and present their responses to three questions in the table, and the diagnostic reports in the subfigures. (For conciseness, we only present part of the responses and diagnositc reports.) From the figure, we can observe that, both IRT and NeuralCDM provides point-wise proficiencies of students. For U-IRT, similar proficiencies are reported (i.e., Student 2 < Student 1 < Student 3); For U-NeuralCDM, the modes of the contributions are also close to the results of NeuralCDM (e.g., the proficiency on F is around 0.65). What's more, U-IRT and U-NeuralCDM provide the uncertainty of their diagnostic results. For example, in Figure 6(d), U-NeuralCDM is quite confident in C (having the most related responses), but more uncertain on B. Based on the uncertainty information, users (e.g., teachers) can decide whether to assign additional questions for better diagnosis; downstream applications, such as learning materials recommendation, can pay more attention to diagnostic results that are less uncertain. Reducing uncertainty can also be considered in the next-question-selection process in computerized adaptive testing [2].

## 5.7 Impact on Diagnostic Ability (RQ5)

In algorithm designing, it is common to encounter situations where it is difficult to simultaneously satisfy different objectives, requiring a trade-off (e.g, accuracy and efficiency in recommender systems). Ideally, when estimating the uncertainty of CDMs, we do not expect negative impacts on the original diagnostic ability of the CDMs. Therefore, UCD is designed with mild modifications of the original CDM structures in order to smoothly conduct the uncertainty estimation. To validate it, we evaluate the diagnostic performances of

CDMs before and after integrating UCD. Following [31], we use the diagnosed results to predict students' performances on questions in the test set, and use AUC, accuracy as metrics. The results of different models are presented in Table 7. Fortunately, we did not observe such degradation from our method. Moreover, for non-deep learning based U-IRT and U-MIRT, there are considerable improvements, which might benefit from the regularization of prior distributions of diagnostic variables and the gradient descending algorithm.

## 6 CONCLUSION

In this paper, we proposed a unified solution to the uncertainty estimation of cognitive diagnosis models (UCD). Compared to traditional approaches, UCD follows the Bayesian strategy but provides better efficiency, and more sufficiently models the differences among parameters into the uncertainty from both data and model aspects. Therefore, UCD can not only be applied to traditional non-deep learning latent trait models but also fill the vacancy for deep learning-based models.

In UCD, we introduced a unified objective function and derived a reparameterization approach that can be applied to large-scale diagnosis model parameters defined on different domains. The current solution is based on the independence assumption among model parameters. In future studies, UCD can be further improved by considering the covariance among diagnostic parameters to better fit advanced cognitive diagnosis models (e.g., KaNCD).

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
