# OpenReview forum: "Unified Uncertainty Estimation for Cognitive Diagnosis Models"
_ACM.org/TheWebConf/2024/Conference — TheWebConf24_

### Official Review · Reviewer_LNAr · 2023-11-18

**Novelty:** 5
**Technical Quality:** 5

**Review:**

Quality：
In this paper, the authors propose a Unified Uncertainty Estimation (UCD) approach for cognitive diagnosis models. This approach is applicable to both traditional latent trait models and deep learning models, filling a gap in the latter. The UCD method encompasses several key components:
1.Unified Objective Function: Based on the concept of learning posterior distributions of parameters, a unified objective function is developed for mini-batch based optimization, suitable for both deep and non-deep learning models.
2.Derivative Reparameterization Approach: This method facilitates efficient gradient descent-based training and is adaptable to parameters with different domains of definition.
3.Differentiation between Diagnostic and Function Parameters: By considering the differences between diagnostic and function parameters, the uncertainty of diagnostic parameters is decomposed into data uncertainty and model uncertainty.
The quality of the paper is high, providing detailed explanations of the methodology, mathematical derivations, and experimental setups. However, some technical details might be challenging for readers unfamiliar with Bayesian methods and cognitive diagnostic models. In terms of quality, UCD shows its versatility and applicability through comprehensive experiments across different cognitive diagnostic models and datasets.
Clarity：
The overall structure of the paper is clear. The research motivation is well-defined, and each section of the paper has a distinct theme. Symbols in formulas are clearly explained, and tables/figures are clear. However, there are some minor errors that could be addressed.
1.	In line 15, the suggestion is to change "limited inefficiency" to "limited efficiency"；
2.	The interpretation of the experimental results on the model aspect in RQ2 is unclear；
3.	Title 5.6's “R4” should be changed to “RQ4”;
4.	 Providing results across different datasets and models would enhance the 	persuasiveness of the findings in RQ4.
Originality：
The innovation of the paper lies in its proposal of a unified solution for uncertainty estimation in cognitive diagnosis models (UCD). This approach is noteworthy for several reasons:
Bayesian Strategy: UCD adopts a Bayesian strategy, aligning with contemporary statistical approaches but enhancing them in specific ways.
Efficiency and Effectiveness: Compared to traditional methods, UCD offers better efficiency. This improvement is crucial in computational models where efficiency can significantly impact practical usability.
Modeling Differences in Uncertainty: The approach more effectively captures the differences in uncertainty stemming from both data and model aspects. This dual focus is essential in accurately reflecting the real-world complexities of cognitive diagnosis.
Applicability: UCD is versatile, applicable not only to traditional non-deep learning latent trait models but also capable of addressing gaps in deep learning-based models.
Overall, the paper's innovation is in its comprehensive and efficient approach to uncertainty estimation, broad applicability, and enhancement of existing Bayesian strategies, however, in certain aspects, such as Bayesian strategy and reparameterization methods, research has already been conducted in related backgrounds, requiring clearer elucidation of their novelty.
Significance：
This paper introduces a novel uncertainty estimation method for cognitive diagnostic models, referred to as the Unified Certification Solution (UCD). UCD adopts a Bayesian strategy and efficiently estimates uncertainty in both traditional and deep learning-based cognitive diagnostic models through the introduction of a unified objective function and a reparameterization method. The method is extensively validated across various cognitive diagnostic models and multiple datasets, demonstrating its robustness and reliability.
The significance of the paper lies in addressing a crucial challenge in the field of cognitive diagnosis by proposing a universally applicable method that effectively estimates uncertainty in different models and datasets.
In conclusion, this paper provides a high-quality research contribution, offering a novel and comprehensive solution to the uncertainty estimation problem in cognitive diagnostic models.
Pros:
1.	The overall structure of the paper is clear, providing detailed explanations of the methodology, mathematical derivations, and experimental setups.
2.	This paper proposes a unified solution to the uncertainty estimation of cognitive diagnosis models, provides better efficiency.
3.	The author conducted a series of comprehensive experiments to validate the effectiveness of the proposed method.
4.	UCD is not only applicable to traditional non-deep learning latent trait models but also suitable for deep learning models, filling a gap in this field.
Cons:
1.	There may be some aspects in the explanation of certain charts and formulas that require clearer clarification to ensure that readers can accurately comprehend the author's points.
2.	The research of Bayesian strategy and reparameterization methods has already been conducted in related backgrounds, this paper may lack a certain degree of innovation.
3.	The author's explanation of the experimental results regarding the model's uncertainty is not sufficiently clear.
4.	The experiments regarding RQ4 could be further enriched.

**Questions:**

1.	Could you provide a clearer explanation of the model uncertainty?
2.	In RQ2, I do not understand the experimental results of the model aspect. What kind of correlation exists between the model and the parameters?

**Reviewer Confidence:**

3: The reviewer is confident but not certain that the evaluation is correct

**Scope:**

4: The work is relevant to the Web and to the track, and is of broad interest to the community

---

### Official Review · Reviewer_4Aad · 2023-11-19

**Novelty:** 3
**Technical Quality:** 4

**Review:**

This paper address important issue of uncertainty estimation in deep learning based Cognitive Diagnosis Models.

Paper proposes to use mini-batch based optimization and reparameterization trick.

The proposed approach is compared with existing uncertainty estimation methods in Cognitive Diagnosis Models.

Pros
The novelty of paper is in application of the approach for Cognitive Diagnosis Models

Cons
The proposed approach of uncertainty estimation has been used previously in variational inference.

**Questions:**

1. Could you describe novelty of the uncetainty estimation method itself vs novelty of application of existing method to a new area  of Cognitive Diagnosis Models.
2. Could you assess the accuracy of estimated uncertainty with proposed method and compare to existing methods

**Ethics Review Description:**

no ethics concerns

**Reviewer Confidence:**

2: The reviewer is willing to defend the evaluation, but it is likely that the reviewer did not understand parts of the paper

**Scope:**

3: The work is somewhat relevant to the Web and to the track, and is of narrow interest to a sub-community

---

### Official Review · Reviewer_anbH · 2023-11-23

**Novelty:** 4
**Technical Quality:** 5

**Review:**

This work presents an approach called Unified Uncertainty Estimation for Cognitive Diagnosis Models (UCD) to estimate the uncertainty of measurement in cognitive diagnosis models. UCD proposes a unified objective function for mini-batch based optimization and modifies the reparameterization approach. The uncertainty of diagnostic parameters is divided into two aspects, data aspect and model aspect, for better explainability. The authors verify their method with comprehensive experiments.

Pros:
1. Better efficiency provided by UCD with good generalizability.
2. Detailed descriptions on proposed method with formulas and pseudo codes.

**Questions:**

-

**Reviewer Confidence:**

1: The reviewer's evaluation is an educated guess

**Scope:**

3: The work is somewhat relevant to the Web and to the track, and is of narrow interest to a sub-community

---

### Official Review · Reviewer_HApC · 2023-11-27

**Novelty:** 6
**Technical Quality:** 5

**Review:**

Summary:
The paper presents a unified approach to uncertainty estimation for cognitive diagnosis models (CDMs), commonly used in intelligent education to assess user proficiency levels. This approach addresses the challenge of unreliable measurements in CDMs by introducing a batch-based optimization method applicable to various models and large datasets. It modifies the reparameterization approach for better adaptation to parameters defined in different domains and decomposes the uncertainty of diagnostic parameters into data and model aspects, enabling a more accurate and reliable assessment of user proficiency levels​.

Strengths:
1. The paper introduces a novel method for estimating uncertainty in CDMs, a critical aspect often overlooked in traditional approaches.
2. The proposed method's compatibility with a wide range of models and its efficiency in handling large datasets make it highly versatile and applicable in diverse educational settings.
3. By decomposing uncertainty into data and model aspects, the paper offers a more nuanced understanding of the sources of uncertainty, leading to more reliable proficiency assessments.
4. The modification of the reparameterization approach to suit parameters across different domains enhances the model's adaptability and accuracy.

Weaknesses:
1. The proposed method's complexity, especially in terms of uncertainty decomposition and reparameterization, might make it challenging to implement and understand for practitioners new to the field.
2. The approach might be prone to overfitting, especially when applied to highly specific or limited datasets.
3. The paper does not explicitly address the method's generalizability across different educational settings or subject matters.

**Questions:**

Could you provide insights into the scalability of your approach in various educational settings, particularly those with limited resources?

**Reviewer Confidence:**

3: The reviewer is confident but not certain that the evaluation is correct

**Scope:**

4: The work is relevant to the Web and to the track, and is of broad interest to the community

---

### Official Review · Reviewer_4DEH · 2023-11-30

**Novelty:** 5
**Technical Quality:** 6

**Review:**

Cognitive diagnostic models have found widespread application across different domains. However, the research on estimating model uncertainty still faces several limitations, including 1) a limited range of applicable algorithms, 2) inadequate parameter analysis, and 3) low efficiency in uncertainty estimation. To overcome these challenges, this study introduces a unified uncertainty estimation method called the Uncertainty Estimation Approach for Cognitive Diagnosis models (UCD), which can be applied to diverse cognitive diagnostic models. In contrast to traditional approaches, UCD adopts a Bayesian strategy that offers improved efficiency and effectively captures parameter differences as uncertainty from both data and model perspectives. Consequently, UCD is not only suitable for traditional non-deep learning latent trait models but also fills the gap in uncertainty estimation for deep learning-based models.

Pros:
1. The paper exhibits good organization and clarity in its presentation, resulting in excellent readability.
2. The examples and case studies used in the paper clearly illustrate the problem and its practical significance (e.g., Figure 1 and Figure 6).
3. The topic addressed is intriguing and relevant to current industry needs. The proposed method is applicable to multiple models and different problems.

Cons:
1. There is a need to adjust the font sizes of certain figures in the paper to align them with the font size used in the main text. Specifically, the fonts in Fig. 2 and Fig. 3 are oversized, while the font in Fig. 5 appears to be undersized.
2. The authors primarily provided the mean results in the experiments, and it is requested that they also provide information about the variance of the results.
3. In some tables, it would be beneficial to highlight the better results by bolding them for easier comparison.

**Questions:**

1. What is the variance information of the experimental results?

**Reviewer Confidence:**

3: The reviewer is confident but not certain that the evaluation is correct

**Scope:**

4: The work is relevant to the Web and to the track, and is of broad interest to the community

---

### Decision · Program_Chairs · 2024-01-22

**Decision:**

Accept

**Comment:**

While I think the paper is highly relevant to the UMAP track and I personally enjoy a hierarchical Bayesian (deep) model for psychometric modeling, the reviews (both explicitly stated and implied in terms of what many reviewers commented on) suggest that the work's impact on TheWebConf may be somewhat narrowly focused on a small audience.

 Since most reviewer concerns are not deeply technical so I felt obligated to read the paper myself given that it's final review scores leave it borderline.

 Having read through the paper, I note that it could use a pass for grammatical (use of articles, prepositions, noun-verb agreement) and technical terminology improvement (e.g., the algorithm stands for Expectation Maximization, not Expectation Maximum). These are minor and infrequent and thus did not disrupt my reading, but they should be fixed on revision.

 At first glance, as one reviewer notes, the paper does seem to reflect a variational Bayesian approach to posterior estimation through minimizing KL divergence. But as the authors note in their response, their approach is much more customized for Cognitive Diagnosis through it's decomposition of uncertainty parameters. So I do believe that the overall novelty of this paper is quite high. Technical clarifications raised by reviewers can be easily fixed on revision.

 My primary concern is that for this tool to be useful, we must be able to run it easily and reliably without manual tuning on test data. No details of the training process are provided beyond Algorithm 1 which is the core optimization routine for the training data. What concerns me is that there does not appear to be any specific discussion of how the various hyperparameter (learning rate, M_c), optimizer choice (SGD, Adam), and the stopping criteria are determined, nor whether held-out validation data was used to determine any of these choices or hyperparameters. The authors provide code, which presumably would answer these questions for anyone willing to read the code, but it is critical for the methodology to be explained in the paper (or an Appendix) in order for readers to understand how this approach is applied in practice to reproduce the experimental results.

 Due to the above reasons, I think the paper remains on the borderline going into the final decision. It has high novelty though a somewhat narrow scope of impact. Further, the paper would significantly benefit from an Appendix describing details on the training methodology to understand how key training and hyperparameter choices should be made by a practitioner using this fairly complex methodology.